# FLT3 Tyrosine Kinase Inhibitors for the Treatment of Fit and Unfit Patients with FLT3-Mutated AML: A Systematic Review

**DOI:** 10.3390/ijms22115873

**Published:** 2021-05-30

**Authors:** Michael Loschi, Rinzine Sammut, Edmond Chiche, Thomas Cluzeau

**Affiliations:** 1Service d’Hématologie Clinique, Centre Hospitalier Universitaire de Nice, 06200 Nice, France; chiche.e@chu-nice.fr (E.C.); rinzine.sammut@chu-nice.fr (R.S.); cluzeau.t@chu-nice.fr (T.C.); 2Université Cote d’Azur, 06108 Nice, France; 3Centre Méditerranéen de Médecine Moléculaire, 06204 Nice, France

**Keywords:** FMS-like tyrosine kinase 3 (FLT3), acute myeloid leukemia, intensive chemotherapy, demethylating agent, remission, survival, relapse, allogeneic stem cell transplantation, maintenance

## Abstract

FLT3-mutated acute myeloid leukemia accounts for around 30% of acute myeloid leukemia (AML). The mutation carried a poor prognosis until the rise of tyrosine kinase inhibitors (TKIs). New potent and specific inhibitors have successfully altered the course of the disease, increasing the complete response rate and the survival of patients with FLT3-mutated AML. The aim of this article is to review all the current knowledge on these game-changing drugs as well as the unsolved issues raised by their use for fit and unfit FLT3-mutated AML patients. To this end, we analyzed the results of phase I, II, III clinical trials evaluating FLT3-TKI both in the first-line, relapse monotherapy or in combination referenced in the PubMed, the American Society of Hematology, the European Hematology Association, and the Clinicaltrials.gov databases, as well as basic science reports on TKI resistance from the same databases. The review follows a chronological presentation of the different trials that allowed the development of first- and second-generation TKI and ends with a review of the current lines of evidence on leukemic blasts resistance mechanisms that allow them to escape TKI.

## 1. Introduction

Acute myeloid leukemia (AML) is the most common type of leukemia in adults [1,2]. It is a heterogeneous group of hematological malignancies characterized by the clonal proliferation of immature myeloid cells in the bone marrow [1]. In its most recent classification, the World Health Organization identified AML with recurrent genetic abnormalities, AML with myelodysplasia-related changes, therapy-related myeloid neoplasms, and AML not otherwise specified [3]. The incidence increases with age, with 50% of the patients being older than 60 years old. Treatment of AML for younger and fit patients usually relies on chemotherapy. For older patients, the new standard of treatment is a combination of hypomethylating agents and venetoclax. In younger patients, the first round of chemotherapy usually yields a complete remission rate of 60 to 70%. The success of the AML treatment is mostly dependent on recurrent cytogenetic and genetic alterations. The 2017 European Leukemia Net classification has divided AML patients into three prognosis groups based on their oncogenetic characteristics [4]. 

Among these recurrent genetic alterations, the mutations or duplication of the FMS such as tyrosine kinase 3 (FLT3) is the most frequent. 

FLT3 is a member of the class III receptor tyrosine kinase family that exerts a key role in the proliferation and differentiation of hematopoietic progenitor cells. Around 30% of newly diagnosed AML patients carry a genetic modification in the *FLT3* gene [5]. It can either affect the juxtamembrane domain and the activation loop of the tyrosine kinase domain (TKD) (~7% of AML cases), resulting in the constitutive activation of FLT3, or it can be an internal tandem duplication (ITD), leading to a disrupted juxtamembrane domain that has been shown to be crucial for kinase autoinhibition (~23% of AML cases) [6]. These activated mutations in *FLT3* are oncogenic and show transformation activity in cells. Before the rise of small molecule inhibitors, FLT3-mutated AML was associated with a poor prognosis with a high relapse risk [7,8,9] even after allogeneic hematopoietic stem cell transplantation (aHSCT) [10]. Many studies have now reported a link between the rate of ITD allelic ratio and prognosis, with a ratio higher than 0.5 associated with a higher risk of treatment failure and relapse [11]. The prognostic impact of *FLT3* TKD mutation is still debated. The real breakthrough for these patients came with the use of FLT3 inhibitors either in monotherapy or in combination with standard treatment. In the article, we will review the current therapeutic options for both fit and unfit patients with FLT3-mutated AML.

## 2. Pharmacological Inhibition of FLT3 in AML

### 2.1. TKI in the Treatment of AML Patients 

The poor prognosis associated with FLT3-mutated AML has led to the development of pharmacological agents designed to inhibit FLT3. The development of FLT3-targeting drugs proved to be difficult initially as many of the available candidates had poor bioavailability, low potency, insufficient kinase specificity, leading to few responders [12]. Despite these setbacks, several FLT3 inhibitors have passed the preclinical development requirements and have been evaluated in clinical trials. Indeed, many improvements have been achieved with next-generation TKIs, as compared to sorafenib and Lestaurtinib. Past and present TKI especially have different spectra and do not inhibit the same kinases (Table 1). These differences partly explain the drugs’ side effects and efficacy. Current FLT3 inhibitors are usually divided between first- (Sorafenib, Midostaurin, and Lestaurtinib) and second-generation TKIs (gilteritinib, quizartinib, crenolanib). Next-generation TKIs have fewer off-target effects leading to fewer and less severe side effects and a higher FLT3 specificity [13,14]. Most common side effects now include cytopenias and gastrointestinal effects such as nausea and vomiting. These toxicities are easily manageable through transfusions, antibiotic courses, and antiemetics. A brief overview of the different developmental phase of the *FLT3* TKI is displayed in Table 2, whereas a summary of the clinical trials that led to the development of the most recent TKIs is provided in Table 3.

#### 2.1.1. Midostaurin

Midostaurin, also known as PKC412 or CGP41251, is an orally administered inhibitor. Its kinome includes the wild-type (WT) and mutated FLT3, KIT, VEGFR, and PDGFR. After binding its target, Midostaurin inhibits the signaling pathways regulated by the kinase leading to a growth arrest. The drug is active on both *FLT3*-ITD and *FLT3*-TKD derived kinase. Its metabolism goes through CYP3A4, which may result in several drug interactions.

Midostaurin was first assessed in a phase II study of 20 relapsed/refractory FLT3-mutated AML patients who received 75 mg Midostaurin three times daily. No patients achieved complete remission (CR), but the drug showed some promise by reducing both the bone marrow and the peripheral blast cell counts in, respectively, 50% and 70% of the patients [15]. 

The inhibitor was then tested during a phase II trial in patients with R/R AML or high-risk MDS with WT or mutated FLT3. Patients received single-agent Midostaurin at a dose of 50 mg BID or 100 mg BID. There was no difference in terms of tolerability between the two dosage groups. Again, none of the patients achieved CR but the blast count decreased in 71% of the FLT3-mutated AML patients and 42% of the nonmutant patients [16].

A subsequent phase Ib trial was then designed [17]. This study enrolled adult patients, aged 18 to 60 years old, with untreated AML. Patients received several schemes of Midostaurin administration in combination with chemotherapy, including 100 mg BID and 50 mg BID, continuous administration or from day 8 to day 21. The sequential treatment from day 8 to 21 was the best-tolerated regimen and did not yield severe gastrointestinal or liver complications. Regarding efficacy, 80% of all participants achieved complete remission (74% in the WT group and 92% in the FLT3-mutated group). In the FLT3-mutated group, the 1- and 2-year overall survival were, respectively, 85% and 62%.

These encouraging data led to the phase III, randomized, placebo-controlled RATIFY trial [18]. It enrolled newly diagnosed *FLT3* ITD or TKD AML patients. The primary endpoint of the study was overall survival. Secondary key endpoints were event-free survival, complete remission (CR), disease-free survival (DFS), and pharmacokinetics data. The study was designed to show that the combination of Midostaurin and conventional induction and consolidation chemotherapy, followed by one-year maintenance, would improve the OS of patients with FLT3-mutated patients. Of note, patients were not allowed to receive posttransplant Midostaurin maintenance in this trial. 

The study included 717 patients, 360 in the Midostaurin arm and 357 in the placebo group. The *FLT3* subtype was ITD for 555 patients and TKD for 162 patients. The study achieved its primary endpoint by demonstrating a significantly longer survival for patients receiving Midostaurin (*p* = 0.0078). The probability of being alive at 3 years was higher in the Midostaurin group, compared to the placebo group (54% [95% CI: 0.49, 0.59] vs. 47% [95% CI: 0.41, 0.52]). There was no difference in the frequency of hematological and nonhematological adverse events. No difference was noted regarding CR rate 59% in the Midostaurin group and 54% in the placebo group (*p* = 0.15). The 4-year EFS rates were 28.2% in the Midostaurin group and 20.6% in the placebo group. The 4-year OS rates were 51.4% in the Midostaurin arm versus 44.3% in the placebo arm. The hazard ratio (HR) comparing Midostaurin to placebo was 0.78 for OS (one-sided *p* = 0.009) and 0.78 for EFS (one-sided *p* = 0.002). All *FLT3* subgroups (ITD and TKD) benefited from the addition of Midostaurin.

The RATIFY trial was the first time a new molecule ever demonstrated a real benefit in addition to chemo for first-line treatment of fit and young AML patients. This trial led to the approval of the drug by both the EMA and the FDA for the first-line treatment of adult patients with FLT3-mutated AML, in combination with the standard 7 + 3 regimen [19]. The EMA also granted Midostaurin authorization for maintenance therapy after complete remission. Following the RATIFY trial, the results of a phase 3b trial were reported at the ASH meeting in 2020 [20]. The aim of the study was to evaluate further the safety and efficacy of the combination of Midostaurin with chemotherapy both in younger and older patients with FLT3-mutated AML. This European multicenter open-label study enrolled 300 patients and confirmed that the combination of Midostaurin plus chemotherapy is safe and results in high response rates regardless of patient age or induction regimen, with a CR/CRi rate of 80.7%. 

#### 2.1.2. Lestaurtinib

Lestaurtinib is another FLT3 inhibitor candidate that was studied in two randomized trials the UK AML15 and AML17 trials, comparing the outcome of newly diagnosed FLT3-mutated AML patients with or without the addition of Lestaurtinib to conventional chemotherapy [21]. The patients were started on Lestaurtinib or placebo 2 days after the completion of their induction and consolidation courses for a 28-day period. No significant differences were seen in either 5-year overall survival or 5-year relapse-free survival. Subgroups’ analysis demonstrated large interpatient differences in FLT3 inhibition, with a survival advantage for patients with a stronger rate of inhibition.

Aside from these first-generation TKIs, several second-generation have been developed, among which Gilteritinib and Crenolanib are type I inhibitors that target both the inactive and active conformational states of the FLT3 kinase domain, whereas Quizartinib is a type II inhibitor that is specific for the inactive conformation [22]. These drugs were developed to have a better FLT3 affinity and specificity. 

#### 2.1.3. Quizartinib 

Quizartinib is a once-daily, oral, highly potent, and selective, second-generation, type II FLT3 inhibitor that has shown antitumor activity in *FLT3*-ITD acute myeloid leukemia in animal models [23]. Its spectrum includes cKIT, another type III receptor tyrosine kinase. This special spectrum characteristic explains Quizartinib’s increased bone marrow toxicity. Quizartinib is not active on *FLT3* TKD AML.

Quizartinib clinical results were first evaluated in a phase I trial of Quizartinib monotherapy in R/R AML patients [24]. In this trial, the drug was administered orally as a single dose. Out of 76 patients included in the study, responses occurred in 23 (30%), including 10 (13%) CR and 13 (17%) partial remissions (PRs). The response rate was higher among patients with *FLT3*-ITD-positive patients, with a 53% overall response rate. The median duration of response was 13.3 weeks; the median survival was 14.0 weeks. The drug regimen was well tolerated since few patients exhibited adverse events (10%). The most common drug-related adverse events were digestive (nausea, vomiting, dysgeusia) and prolonged QT interval.

Following this trial, two phase 2 trials investigated Quizartinib in R/R FLT3-mutated AML patients. In an open-label, multicenter, single-arm, phase 2 trial, 333 patients were divided into two independent cohorts: patients who were aged 60 years or older with relapsed or refractory acute myeloid leukemia within 1 year after first-line therapy (cohort 1) and those who were 18 years or older with relapsed or refractory disease following salvage chemotherapy or hemopoietic stem cell transplantation (cohort 2). The co-primary endpoints were the proportion of patients who achieved a composite complete remission (defined as complete remission + complete remission with incomplete platelet recovery + complete remission with incomplete hematological recovery) and the proportion of patients who achieved a complete remission. Tolerability was also recorded. A total of 56% *FLT3*-ITD-positive patients and 36% *FLT3*-ITD-negative patients achieved composite complete remission, with three (3%) *FLT3*-ITD-positive patients and two (5%) *FLT3*-ITD-negative patients achieving complete remission in cohort 1. In cohort 2, 46% of *FLT3*-ITD-positive patients achieved composite complete remission with five (4%) achieving complete remission, whereas 30% of *FLT3*-ITD-negative patients achieved composite complete remission with one (3%) achieving complete remission. The drug exhibited few adverse events, mostly linked to myelosuppression and QT prolonged interval, already described during the phase I trial.

Another phase IIb trial evaluated the safety and efficacy of two regimens of Quizartinib monotherapy in patients with relapsed/refractory (R/R) *FLT3*-ITD AML who previously underwent transplant or 1 s-line salvage therapy. A total of 76 patients were randomly assigned to receive either 30 or 60 mg/day doses (escalations to 60 or 90 mg/day, respectively, permitted for lack/loss of response) of single-agent oral Quizartinib. Similar to the other phase 2 study, co-primary endpoints were composite complete remission (CRc) rates but also the incidence of QT interval. CRc rates were 47% in both groups, similar to earlier reports with higher Quizartinib doses. Incidence of QTcF above 480 ms was 11% and 17%, and QTcF above 500 ms was 5% and 3% in the 30 and 60 mg groups, respectively. The median OS in this trial were 20.9 and 27.3 weeks in the 30 mg group and 60 mg group, respectively.

Following these good results on efficacy and tolerability, Quizartinib was assessed in the QuANTUM-R multicenter, randomized, controlled, open-label, phase 3, comparing Quizartinib versus salvage chemotherapy [25]. The study randomized a total of 367 R/R *FLT3*-ITD patients. Overall survival was longer for Quizartinib than for chemotherapy (hazard ratio 0.76 [95% CI 0.58–0.98; *p* = 0.02]). Median overall survival was 6.2 months (5.3–7.2) in the Quizartinib group and 4.7 months (4.0–5.5) in the chemotherapy group. The most common toxicity in the Quizartinib group was hematological, whereas, in this trial, very few prolonged QT were recorded.

Aside from its use in the R/R AML setting, Quizartinib has been evaluated in combination with intensive chemotherapy during induction and consolidation courses, in a phase 1, open-label, sequential group dose-escalation trial [26]. This study was the first to evaluate the safety and tolerability of Quizartinib in combination with chemotherapy in newly diagnosed AML. Starting on day 4 after chemotherapy, 19 patients received Quizartinib. The tolerability was not different from the one already reported in the previous trials with only one case of the prolonged QT interval. In total, 16 patients (84%) achieved a response, from which 14 (74%) achieved composite complete responses. Following the publication of the QUANTUM R results, a retrospective French multicenter study reported the efficacy and safety of Quizartinib in R/R FLT3-mutated patients enrolled in the Quizartinib early access program [27]. Patients received Quizartinib monotherapy (83%) or in combination with either azacitidine (13%) or intensive chemotherapy (4%). The data from this study revealed that Quizartinib was safely administered with only one case of prolonged QTc and 7% of the patients experiencing differentiation syndrome. Regarding the outcome, the data confirmed the efficacy of Quizartinib, with 15 patients achieving CR or CRi (26%), 2 patients partial response (PR) (4%), 1 patient morphologic leukemia-free state (MLFS) (2%), 9 patients hematological improvement (HI)(16%), or 17 patients in stable disease (30%). These interesting results on R/R need to be confirmed on a larger real-life cohort and in first-line therapy. 

The combination of Quizartinib with chemotherapy as a first-line treatment for FLT3-mutated AML patients is currently under investigation in the QuANTUM-First phase 3 trial [28]. If the results of this trial will change the current standard of care chemotherapy plus Midostaurin remains to be seen as the control arm of the QuANTUM-First trial used a placebo instead of Midostaurin, making it harder to draw any conclusion on the ability of Quizartinib to challenge Midostaurin in first-line treatment in combination with chemotherapy.

#### 2.1.4. Gilteritinib

Gilteritinib is a second-generation TKI, and its spectrum includes FLT3, AXL, and ALK. Several in vitro studies have highlighted the pharmaceutical properties of Gilteritinib. A central point is that the drug targets both *FLT3*-ITD and TKD and has a strong affinity for FLT3, leading to sustained FLT3 inhibition [29]. Since it has no effect on c-kit, Gilteritinib does not yield myelosuppression as much as other second-generation TKIs. Moreover, the AXL inhibition leads to better control of the AML oncogenic pathways and decreases the risk of resistance [30].

The results of the open-label, first-in-human phase I/II trial were recently reported. A total number of 252 R/R AML patients were enrolled into one of seven dose-escalation (*n* = 23) or dose-expansion cohorts (*n* = 229) assigned to receive once-daily doses of oral Gilteritinib (20 mg, 40 mg, 80 mg, 120 mg, 200 mg, 300 mg or 450 mg). The primary endpoints were the safety, tolerability, and pharmacokinetic profile of Gilteritinib. Secondary endpoints included overall response, duration of response, and overall survival [31].

Overall, tolerance was good. The most frequent adverse effects were infections (febrile neutropenia, pneumonia, sepsis), anemia, and thrombopenia. Regarding the efficacy of Gilteritinib monotherapy in this R/R AML population, the ORR rate was 40%, with 19 patients achieving CR, 46 patients achieving complete remission with incomplete hematologic recovery, and 25 partial remissions. The median duration of response was 17 weeks, and the median overall survival was 25 weeks. 

These results paved the way for further trials. Recently reported multicenter, phase 3 ADMIRAL trial reported the outcome of 247 R/R patients assigned to either Gilteritinib monotherapy (at a dose of 120 mg per day) or salvage chemotherapy at a 2:1 ratio [32]. Patients were required to have *FLT3*-ITD or TKD D835 or I836 mutations. The two primary endpoints were overall survival and the percentage of patients who had complete remission with full or partial hematologic recovery. Key secondary endpoints were event-free survival and the percentage of patients with complete remission. Regarding tolerance, the trial reported the same rate and type of adverse events, especially febrile neutropenia, anemia, and thrombopenia, as previously reported in the phase I/II trial. Of note, adverse events were less frequent in the Gilteritinib group, as compared to the salvage chemotherapy group. The median overall survival in the Gilteritinib group was significantly longer than that in the chemotherapy group (9.3 months vs. 5.6 months; hazard ratio for death, 0.64; 95% confidence interval [CI], 0.49 to 0.83; *p* < 0.001). The median event-free survival was 2.8 months in the Gilteritinib group and 0.7 months in the chemotherapy group (hazard ratio for treatment failure or death, 0.79; 95% CI, 0.58 to 1.09). The percentage of patients who had complete remission with full or partial hematologic recovery was 34.0% in the Gilteritinib group and 15.3% in the chemotherapy group (risk difference, 18.6 percentage points; 95% CI, 9.8 to 27.4); the percentages with complete remission were 21.1% and 10.5%, respectively (risk difference, 10.6 percentage points; 95% CI, 2.8 to 18.4). 

Based on these encouraging results, several phase 3 trials have been designed to evaluate the safety and efficacy of Gilteritinib in combination with conventional chemotherapy, compared to Midostaurin (in Europe: HOVON 156 AML/AMLSG 28-18, NCT04027309, in the USA: NCT03836209)

#### 2.1.5. Crenolanib

Just as Gilteritinib, Crenolanib is a type II inhibitor. Crenolanib has a heightened binding affinity against the *FLT3*-ITD mutation (the most frequent *FLT3* aberration in AML) and TKD point mutations *FLT3* (D835H), *FLT3* (D835Y), and *FLT3* (D835V). Interestingly, Crenolanib has the ability to suppress all resistance-conferring TKD mutants [33]. 

Similar to other FLT3 ITKs, Crenolanib has first been studied among R/R FLT3 mutant patients. In the ARO-010 phase I/II trial, 28 heavily pretreated patients were included and received salvage therapy with Crenolanib. Crenolanib was well tolerated at a dose of 100 mg TID in combination with salvage chemotherapy. The main reported side effects were nausea, diarrhea, constipation, and fatigue [34]. A second multicenter phase I trial ARO-011 (NCT02626338) confirmed these results on 16 R/R FLT3-WT and -mutated AML patients [35]. Reported in 2019, the results of another phase II trial reported that Crenolanib when given with chemotherapy can achieve sufficient levels to inhibit multiple FLT3 mutations. The combination of Crenolanib and salvage chemotherapy seemed to improve outcomes in younger patients with newly diagnosed FLT3-mutated AML. 

Crenolanib is currently studied in two multicenter, international, randomized trials, both in R/R AML patients (NCT03250338) [36] and in newly diagnosed FLT3-mutated AML (NCT03258931) [37], compared to Midostaurin.

**Table 3 ijms-22-05873-t003:** Summary of TKIs’ main clinical trials results.

Reference	Drug	Study Design	Number of Enrolled Patients	Treatment Phase	Type of FLT3 Mutant	Primary Endpoint	Survival Rates	Toxicities
Stone et al. [16]	Midostaurin	Phase 2	20	R/R FLT3 mut AML	*FLT3-ITD* and *TKD*	Antileukemic activity	N/A	Nausea/vomiting
Fischer et al. [17]	Midostaurin	Phase 2	95	R/R AML 35 FLT3 mut	*FLT3-ITD* and *TKD*	Safety/tolorability	N/A	
				60 FLT3 WT				
Stone et al. [19]	Midostaurin	Phase 3	717	First line	*FLT3-ITD* and *TKD*	OS	OS midostauin > OS placebo (*p* = 0.009)	Anemia (more frequent in the Midostaurin group)
Cortes et al. [25]	Quizartinib	Phase 1	76	R/R AML	Irrespective of *FLT3-ITD status*	Tolerability	N/A	Nausea, prolonged QT interval, vomiting, dysgeusia
Cortes et al. [26]	Quizartinib	Phase 3	367	R/R AML	*FLT3-ITD*	OS	OS Quizartinib > OS salvage chemotherapy (*p* = 0.02)	Nausea, prolonged QT interval, vomiting, dysgueusi
Perl et al. [32]	Gilteritinib	Phase 1/2	252	R/R AML	*FLT3 mut* and *WT*	Tolerability, safety, pharmacokinetics	N/A	Infections/Anemia/thrombopenia
Perl et al. [33]	*Gilteritinib*	*Phase 3*	371	R/R AML	*FLT3-ITD* and *TKD*	OS and CR rates	OS Gilteritinib > OS salvage chemotherapy (*p* < 0.001). CI, 0.58 to 1.09)	Febrile neutropenia/anemia/thrombopenia
Aboudalle et al. [35]	Crenolanib	Phase I/II	28	R/R AML	*FLT3-ITD* and *TKD*	Tolerability/overall response rate (ORR)	N/A	No dose-limiting toxicity/no death related to Crenolanib/ORR: 46%

### 2.2. FLT3 TKI and Demethylating Agents

Aside from their association with intensive chemotherapy in fit patients, FLT3 TKIs have also been studied in unfit patients ineligible for intensive chemotherapy (Table 4). The trials conducted on these combinations have enrolled small cohorts. The first studies have used first-generation Sorafenib. The combination of Sorafenib and Azacitidine in 43 unfit patients with R/R AML yielded a complete remission rate of 16% with a median OS of 6.2 months in a prospective phase II trial [38]. In a retrospective study, the outcome of 26 *FLT3*-ITD AML patients older than 60 years of age treated with a combination of Sorafenib and Azacitidine was compared to a historical cohort of patients treated with Azacitidine only [39]. In this study, there were no differences between the two groups regarding ORR and OS. The combination of Sorafenib and Azacitidine was safe and efficient. Another study reported the outcome of six patients treated with a combination of Sorafenib and another demethylating agent, Decitabine. The combination was well tolerated, and four patients achieved complete remission.

Second-generation TKI Midostaurin was evaluated in combination with Decitabine for adult AML patients with relapsed or refractory (R/R) disease, or elderly (≥60 years) patients unable to receive standard induction chemotherapy. The combination was well tolerated but resulted in few CR [40]. A Phase I/II trial later assessed the efficacy of the combination of Midostaurin with Azacitidine in 54 unfit patients with high-risk myelodysplastic syndromes (MDS) or AML. The overall response rate was 26%, with a median remission duration of 20 weeks. Of note, the remission was significantly longer for the patients with FLT3 mutations not previously exposed to FLT3 inhibitors [41]. 

A prospective, open-label, single-center phase I/II trial evaluated the safety and efficacy of the combination of Quizartinib with azacitidine (AZA) or low dose cytarabine (LDAC) in patients with R/R FLT3 AML or in first-line treatment for >60 years old patients [42]. Overall, 73 patients were treated (34 front line, 39 first salvage). The study confirmed the safety and efficacy of Quizartinib associated with either low-dose cytarabine or Azacitidine, with a composite response (CRc) rate of 87%, (8 CR, 4 Cri, 1 CRp) among patients treated with Quizartinib/AZA and 74% (1 CR, 8 CRi, 5 CRp) in Quizartinib/LDAC. The median OS was 19.2 months for Quizartinib/AZA and 8.5 months for Quizartinib/LDAC cohort. Only one patient experienced a prolonged QTc. Despite encouraging preliminary results [44], the phase 3 LACEWING trial of the FLT3 inhibitor Gilteritinib (Xospata^®^, Astellas, Tokyo, Japan) plus Azacitidine versus Azacitidine alone in patients with newly diagnosed FLT3 mutation-positive AML who were ineligible for intensive induction chemotherapy did not meet its primary endpoint of overall survival (OS) at a planned interim analysis, according to Astellas Pharma, the developer of the agent. It was an open-label, multicenter, randomized, phase 3 trial designed to evaluate the use of Gilteritinib plus Azacitidine versus Azacitidine alone in approximately 250 patients with newly diagnosed FLT3 mutation-positive AML who were ineligible for first-line intensive induction chemotherapy (NCT02752035) [43]. 

### 2.3. TKI Maintenance Post Allogeneic Hematopoietic Stem Cell Transplantation

Most second-generation TKIs are currently under investigation in phase III trials after aHSCT for FLT3-mutated AML.

Sorafenib is a multikinase inhibitor that inhibits FLT3 as well as the RAS, RAF, KIT, and the VEGF and platelet-derived growth factor receptor. A first phase 1 trial initially demonstrated that Sorafenib maintenance was safe [45]. During this phase 1 trial, 22 patients received Sorafenib 400 mg BID. Maintenance lasted 12 months. The trial yielded very promising results as for all patients, one-year progression-free survival (PFS) was 85% (90% CI, 66%–94%) and one-year overall survival (OS) was 95% (90% CI, 79%–99%) after aHSCT. Following these promising data, a randomized, placebo-controlled, double-blind phase II trial (the SORMAIN trial), enrolled 83 adult patients with FLT3-ITD-positive AML in complete hematologic remission after HCT who were randomly assigned to receive for 24 months either the multitargeted and FLT3-kinase inhibitor Sorafenib (*n* = 43) or placebo (*n* = 40 placebo) [46]. Relapse-free survival (RFS) was the primary endpoint of this trial. At the end of the study, the hazard ratio (HR) for relapse or death in the Sorafenib group versus placebo group was 0.39 (95% CI, 0.18 to 0.85; log-rank *p* = 0.013). The 24-month RFS probability was 53.3% (95% CI, 0.36 to 0.68) with placebo versus 85.0% (95% CI, 0.70 to 0.93) with Sorafenib (HR, 0.256; 95% CI, 0.10 to 0.65; log-rank *p* = 0.002). In 2020, the first phase III randomized trial eventually demonstrated that patients receiving Sorafenib had a higher overall survival [47]. In an open-label, randomized phase III trial at 7 hospitals in China, 202 patients with an *FLT3*-ITD acute myeloid leukemia, who underwent an aHSCT, were randomly assigned (1:1) to Sorafenib maintenance (400 mg orally twice daily) or non-maintenance (control) at 30–60 days post transplantation. The 1-year cumulative incidence of relapse was significantly lower in the Sorafenib group (7%), as compared to the control group (24.5%) (hazard ratio 0.25, 95% CI 0.11–0.57; *p* = 0.0010). Sorafenib was well tolerated and did not result in an increased rate of side effects. These studies have led to the Sorafenib maintenance recommendation by the European Group for Blood and Marrow Transplantation (EBMT) as posttransplant maintenance. 

Conversely, Midostaurin maintenance following alloHSCT has not proved to improve patient outcomes in the recent phase 2 open-label randomized RADIUS trial of the standard of care (SOC) with or without Midostaurin after alloHSCT for FLT3-mutated AML patients [48]. In this trial, 60 adults (aged 18–70 years old) patients who received alloHSCT in first complete remission, were randomized to receive standard of care with or without Midostaurin (50 mg twice daily) continuously in 12 four-week cycles. The standard of care only included antibioprophylaxis and GVHD prophylaxis and treatment. No additional disease treatment was allowed in the SOC group. The trial did not reach its primary endpoint since RFS was not significantly different between the two groups, i.e., 89% (69–96%) in the Midostaurin arm and 76% (54–88%) in the SOC arm (hazard ratio, 0.46 [95% CI, 0.12–1.86]; *p* = 0.27). The rate of adverse events including GVHD was not different between the two groups. Differences in the kinase spectrum may explain the differences in outcome between Sorafenib and Midostaurin posttransplant.

A multicenter, randomized, placebo-controlled phase III trial evaluating the safety and efficacy of Gilteritinib as post aHSCT maintenance therapy is currently ongoing (NCT02997202) [49].

## 3. Perspectives

FLT3-mutated AML has long been associated with a poor prognosis and a high risk of relapse even after aHSCT. The rise of FLT3 TKIs has dramatically altered the course of the disease with more patients achieving complete remission. For the first time since the 7 + 3, the addition of a new drug has resulted in a better outcome for fit patients with FLT3-mutated AML who can undergo intensive chemotherapy. Several drugs have been developed since first-generation Sorafenib, with better tolerability and higher inhibition of FLT3 kinase activity. Some of these new drugs are currently under investigation in several settings especially in first-line treatment in association with intensive chemotherapy or in maintenance for *FLT3*-ITD or TKD patients after aHSCT. 

In recent years, aside from FLT3 TKI, other drugs have exhibited interesting response rates among patients with FLT3-mutated AML. CPX-351 is a liposomal formulation of cytarabine and daunorubicin approved for the treatment of adults with newly diagnosed, therapy-related acute myeloid leukemia (t-AML) or AML with myelodysplasia-related changes (MRC-AML). In preclinical studies, CPX-351 exhibited potent cytotoxicity against FLT3 AML blasts in vitro [50]. In the first phase 3 trial demonstrating the superiority of CPX-351 over regular 7 + 3 in sAML, there was a trend toward improved survival with CPX-351 versus 7 + 3 in the small subgroup of patients with FLT3-mutated AML [51]. In a retrospective multicenter French study of real-life experience with CPX 351 in the treatment of sAML, the presence of FLT3 mutation did not impact the response to CPX-351 [52]. A phase III trial is currently ongoing to assess the safety and efficacy of the combination of Gilteritinib and CPX-351 for newly diagnosed FLT3 mutated AML patients [53].

Following the results of the VIALE-A trial [54], Venetoclax, a Bcl2 inhibitor, has recently been approved for the treatment of AML patients in combination with AAzacitidine. However, Venetoclax has demonstrated no responses in a small subset of *FLT3*-ITD+ R/R AML mutant patients. Moreover, *FLT3*-ITD mutations emerged at relapse following Venetoclax monotherapy and combination therapy suggesting a potential mechanism of resistance. Recent preclinical data indicate that the combination of Venetoclax with Quizartinib had an improved activity on FLT3-mutated AML cells [55]. Based on these preclinical findings, a phase 2 trial was designed to study the efficacy and safety of the combination DDecitabine, Venetoclax, and FLT3 TKI in front-line treatment with FLT3- mutated AML >60 years and R/R patients >18 years [56]. Patients received DDecitabine 20 mg/m2 IV for 10 days every 4–6 weeks for induction, followed by DDecitabine for 5 days after CR/CRi. Venetoclax dose was 400 mg PO daily. The addition of FLT3i of the clinician’s choice was allowed. In this study, 25 patients with FLT3-mutated AML were treated with the triplet between 30 April 2018 and 10 February 2020. Patients received either Sorafenib, Midostaurin or Gilteritinib. For front-line patients, the CRc rate was 92%, and in R/R AML, the CRc rate was 62% in R/R AML patients. 

Despite these improvements, several issues remain. Among the unanswered questions are how long should patients be treated with these inhibitors and whether all these inhibitors are equally active on all types of FLT3-mutated AML. Another important limitation of FLT3 TKI is the rise of resistant mutant FLT3 clones in patients treated with FLT3 TKIs. For most patients who relapse following FLT3 inhibitors, the relapse is related to the development of drug resistance. Several mechanisms have been described, including, but not limited to, the emergence of clones that are resistant to FLT3 inhibitors being used, but also the protection of leukemia cells by the microenvironment. Regarding clonal evolution, many studies reported the emergence of *FLT3*-TKD leukemic cells in *FLT3*-ITD AML patients treated with FLT3 TKI [57,58]. Many mutation sites have been described along the FLT3 gene affecting leukemic cells’ sensitivity to one or several TKIs [59]. Aside from TKD mutations, non-FLT3-mutated clones have been shown to expand or emerge at relapse in *FLT3*-ITD AML during FLT3 inhibitor treatment. Patients relapsing after TKI inhibitor treatment have undergone extensive genetic sequencing of their leukemic cells. Next-generation sequencing performed in patients relapsing after Gilteritinib or Crenolanib revealed that the relapsing leukemic cells arose from the original *FLT3*-ITD clone or from *FLT3* wild-type subclones displaying several mutations in *TP53*, the RAS pathway (NRAS, KRAS, BRAF, PTPN11, CBL), IDH1/2, ASXL1, or TET2. The emergence of these mutations demonstrated a selection pressure by the FLT3 treatment [60,61]. 

Regarding the duration of FLT3 TKI efficacy after several courses of the drugs, a recent retrospective study reported the outcome of FLT3-mutated AML patients who received sequential courses of FLT3 TKI because of AML relapse [62]. In this trial, 239 patients receiving FLT3 TKI for the first time, either as front-line treatment (56 patients) or as a salvage treatment for an R/R AML (183), were enrolled. The results showed that both in front-line and in salvage treatment, the composite complete response (CRc) rate and OS dropped over time with the successive exposure to FLT3 TKI. In front-line patients treated with an FLT3 TKI, the CRc rates and median overall survival (OS) with the first (*n* = 56), second (*n* = 32), and third FLT3 TKI (*n* = 8) based therapy were 77%, 31%, and 25%, and 16.7 months, 6.0 months, and 1.4 months, respectively. In patients receiving an FLT3 TKI-based therapy for the first time in an R/R AML setting, the CRc rates and median OS were 45%, 21%, and 10%, and 7.9 months, 4.0 months, and 4.1 months with the first (*n* = 183), second (*n* = 89), and third/fourth (*n* = 29) FLT3i-based therapy, respectively. Another important finding of the study is the increased response rate and survival for patients receiving a combination of FLT3 TKI and chemotherapy, as compared to FLT 3 TKI monotherapy during the course of the AML treatment. In front-line cases, CRc rates with single-agent FLT3 TKI versus FLT3 TKI-based combinations in second/third sequential FLT3i exposures were 19% versus 42%, respectively. For patients treated for an R/R AML, the CRc rates with single-agent FLT3 TKI versus FLT3 TKI-based combinations in first FLT3i exposure were 34% versus 53%, respectively, and those with single-agent FLT3 TKI versus FLT3 TKI-based combinations in second/third/fourth sequential FLT3 TKI exposures were 13% versus 25%, respectively. Finally, there is still an unmet need for FLT3-mutated AML unfit patients, ineligible for intensive chemotherapy, since most trials investigating the association of FLT3 TKI with demethylating agents have proven unsuccessful. 

## 4. Conclusions

In the recent era, FLT3 inhibitors have improved the overall survival of patients with previously poor prognoses. These drugs are a major breakthrough in terms of efficacy, carry few adverse effects, and are changing the way we take care of FLT3-mutated AML patients for every step of the treatment, from induction to post-aHSCT maintenance. Some limitations remain for patients unfit for intensive chemotherapy, and there are unresolved questions about the potential clonal selection of resistant mutants in some patients. A summary of the current options for FLT3-mutated AML is provided in Figure 1. 

The above figure has been established using published recommendations. Every patient should, however, be included in a clinical trial.

## Figures and Tables

**Figure 1 ijms-22-05873-f001:**
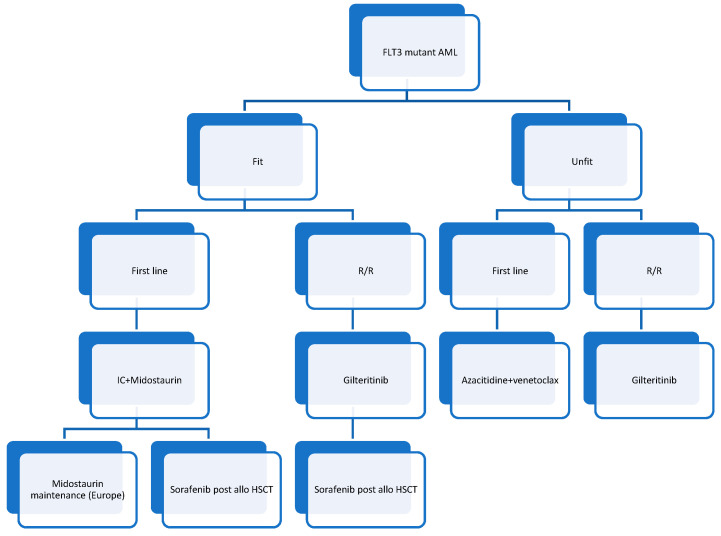
Recommendations for the treatment of fit and unfit patients with FLT3-mutated AML.

**Table 1 ijms-22-05873-t001:** FLT3 TKI spectrum activity.

Agent	Type	Dose	Target
Sorafenib	First generation, type I	400 mg BID	FLT3-ITD, RAF, VEGFR1/2/3, PDGFRβ, KIT, RET
Midostaurin	First generation, type I	50 mg BID	FLT3-ITD, FLT3-TKD, PKC, SYK, FLK-1, AKT, PKA, KIT, FGR, SRC, PDGFRα/β, VEGFR1/2
Quizartinib	Second generation, type II	60 mg once a day	FLT3-ITD, KIT, PDGFR
Gilteritinib	Second generation, type I	120 mg once a day	FLT3-ITD, FLT3-TKD, LTK, ALK, AXL
Crenolanib	Second generation, type I	100 mg TID	FLT3-ITD, FLT3-TKD, PDGFRβ

FMS-like tyrosine kinase 3 (FLT3), rapidly accelerated fibrosarcoma (RAF), vascular endothelial growth factor receptor (VEGFR), JAK Janus kinase (JAK), platelet-derived growth factor receptor (PDGFR), protein kinase (PK), spleen tyrosine kinase (SYK), fetal liver kinase 1 (FLK-1), protein kinase cAMP-dependent (PKA), Gardner–Rasheed feline sarcoma viral (FGR), VEGFR vascular endothelial growth factor receptor.

**Table 2 ijms-22-05873-t002:** FLT3 TKIs. Treatment phase and available trials.

Indication	Agent	Trial
Salvage monotherapy	Midostaurin	Phase I, phase II
	Quizartinib	Phase II, phase III
	Gilteritinib	Phase II, phase III
	Crenolanib	Phase II
Combination with intensive chemotherapy (newly diagnosed)	Midostaurin	Phase III
	Quizartinib	Phase I, Phase III (ongoing)
	Gilteritinib	Phase III (ongoing)
	Crenolanib	Phase III (ongoing)
Combination with demethylating agents	Sorafenib	Retrospective study
	Midostaurin	Phase II study
	Quizartinib	Phase I/II
	Gilteritinib	Phase III (failed)
	Crenolanib	Preclinical
Posttransplantation maintenance	Sorafenib	Phase II, phase III
	Midostaurin	Phase III (failed)
	Quizartinib	Phase III (ongoing)
	Gilteritinib	Phase III (ongoing)
	Crenolanib	Phase III (ongoing)

**Table 4 ijms-22-05873-t004:** FLT3 TKIs in combination with demethylating agents.

Reference	Drug	Study Design	N	Type of FLT3 Mut	Response	Survival
Ravandi et al. [38]	Sorafenib + Azacitidine	Phase II	43	*FLT3-ITD* (93%)	CR 16%	Median OS 6.2 months
Ohanian et al. [39]	Sorafenib + Azacitidine	Phase I/II	27	*FLT3-ITD* (100%)	CR 26%	Median OS 8.3 months
Williams et al. [40]	Midostaurin + Decitabine	Phase I	16	*FLT3-ITD* (13%)	CHR 26%	
Strati et al. [41]	Midostaurin + Azacitidine	Phase I/II	54	*FLT3-ITD* (68%), *TKD* (6%)	ORR 26%	
Swaminathan et al. [42]	Quizartinib + Azacitidine/LDAC	Phase I/II	73	Phase I: *FLT3-ITD* and *WT,* Phase II: *FLT3-ITD* only	Q + A: CR 22% Q + LDAC: CR 8%	Q + A: median OS 13.4 monthsQ + LDAC Median OS: 6.7 months
Wang et al. [43]	Gilteritinib + Azacitidine vs. Azacitidine alone	Phase III	136	FLT3-mutated AML	Trial halted (did not meet endpoint)	

CHR: complete hematologic response; CR: complete remission; LDAC: low-dose cytarabine; ORR: overall response rate.

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
