# Peer review of "FLT3 Tyrosine Kinase Inhibitors for the Treatment of Fit and Unfit Patients with FLT3-Mutated AML: A Systematic Review"

_ijms, 2021, doi:10.3390/ijms22115873_

Round 1

Reviewer 1 Report

In the review manuscript, the authors focused on recent progress on ongoing clinical trials of FLT3 inhibitors in patient with FLT3 mutated acute myeloid leukemia. The manuscript is well written and provide  details of  important clinical ongoing trials in the field of AML treatment.  

There are only few minor comments in my opinion:

- Line 77: 50md BID should read 50 mg BID

- Line 111: Regarding Midostaurin,  the authors indicated that based on ATIFAY trial, the is inhibitor was approved by EMA and FDA as first line of treatment in adult patients with FLT3 AML in combination with 7+3 regiment. However, in the table 1, it is indicated that it is still in phase III. Please revise

- There are also few typo mistakes to be corrected.

Author Response

Dear reviewer thank you for your thorough and encouraging review of our article

We have adressed the issue you have raised

In the review manuscript, the authors focused on recent progress on ongoing clinical trials of FLT3 inhibitors in patient with FLT3 mutated acute myeloid leukemia. The manuscript is well written and provide  details of  important clinical ongoing trials in the field of AML treatment.  

There are only few minor comments in my opinion:

  • Line 77: 50md BID should read 50 mg BID
  • Correction has been made
  • Line 111: Regarding Midostaurin,  the authors indicated that based on ATIFAY trial, the is inhibitor was approved by EMA and FDA as first line of treatment in adult patients with FLT3 AML in combination with 7+3 regiment. However, in the table 1, it is indicated that it is still in phase III. Please revise
  • Correction has been made
  • There are also few typo mistakes to be corrected.
  • we have corrected the typos

Reviewer 2 Report

The authors have omitted to add the Methods section, thus it is difficult to assess this paper. I would like to remind the authors that a systematic review should adhere to the PRISMA guidelines. 

The authors should draw some conclusions based on their findings. It is not sufficient to summarise the results of other studies without drawing some original conclusions. Which FLT3 inhibitors should we use as 1st line? As 2nd line? For fit/unfit patients?

Please add some tables summarising the number of patients in each study, study design, molecular type, outcomes measured, effects of FLT3 inhibitors etc. Please check other published systematic reviews to get an idea.

Author Response

Dear reviewer,

Thank you for your thorough review of our article.

We have addressed the issues you raised in your review.

The authors have omitted to add the Methods section, thus it is difficult to assess this paper. I would like to remind the authors that a systematic review should adhere to the PRISMA guidelines.

For a better reading experience of the manuscript we decided that we would not add a methodology section in the main text of the current review. We have written a Methods section that we would like to add as a supplementary document.

We have corrected the abstract to comply with the PRISMA guidelines.

The authors should draw some conclusions based on their findings. It is not sufficient to summarise the results of other studies without drawing some original conclusions. Which FLT3 inhibitors should we use as 1st line? As 2nd line? For fit/unfit patients?

We have designed a Figure 1 to summarise the current treatment options for FLT3 mutant AML patients

Please add some tables summarising the number of patients in each study, study design, molecular type, outcomes measured, effects of FLT3 inhibitors etc. Please check other published systematic reviews to get an idea.

We have added a table summarising the number of patients in each study, study design, molecular type, outcomes measured, effects of FLT3 inhibitors

Reviewer 3 Report

Line 13- "Knowledge" not "Knowledges"

line 28- can you mention the 3 groups of division before you segue into the next paragraph.

Line 38- use “transformation” instead of “transforming”

Line 54-55  - “proved to be difficult initially as many of the available candidates had poor bioavailability, low potency, insufficient kinase specificity, leading to few responders[12]. De- 55 spite these setbacks”-          could you discuss a bit more how the drugs that have passed pre clinical trials have overcome these obstacles, what are their bioavailability, toxicity or how were these deficiencies addressed?

First letter of  drug brand name should be capitalized, please go through the paper

Line 94- it should be “of note”, instead of “of not”, you are missing the “e”

Line 95- It should be “receive” NOT “received”. Sentence should be, “patients were not allowed to receive

Could you also define or describe what  standard will be considered a “fit” patient.

In line 205 “Quizartinib” first letter was capitalized but it line 187 it was started with a small letter. So there needs to be consistency. In my opinion “brand name” of drugs should be written as a proper noun and started with a capital letter. So you need to go over the paper and do that for all the brand names of the drugs written.

Line 274 it should be  “association with” NOT “association to”. In a in sentence you say that things are “associated with”, or has an “association with”

Line 278, patient treated “with”, preferable use “with” here instead of treated “by”. Same applies to line 282- patients were treated “with” a combination …

Line 308-309 is not very clear, probably need to elaborate on that sentence.

Line333 what is EBMT, at this point the reader would have forgotten what that acronym stands for.

 Line 390- better to use “unanswered question” than “unsolved questions”

Line 431- it should be ineligible “for” intensive…, not “to”. So change the “to” with a “for”

Line 432 -  association of FLT3 TKI “to” demethylating, please change that “to” and use “with”

You say something is associated with, not associated to.

Line 435 this sentence is not complete. Maybe say-  “A major breakthrough about these drugs is that they are associated with minimal adverse events…..”

Line 438 instead of unsolved question either say “unresolved questions” or “unanswered questions”

Author Response

Dear Reviewer,

Thank you for your thorough review and encouraging comments

We have adressed all your comments and made the appropriate corrections.

Line 13- "Knowledge" not "Knowledges"

Correction has been made

line 28- can you mention the 3 groups of division before you segue into the next paragraph.

We have added the groups identified in the 2016 WHO classification : AML with recurrent genetic abnormalities, AML with myelodysplasia related changes, therapy related myeloid neoplasms and AML not otherwise specified

Line 38- use “transformation” instead of “transforming”

Correction has been made

Line 54-55  - “proved to be difficult initially as many of the available candidates had poor bioavailability, low potency, insufficient kinase specificity, leading to few responders[12]. De- 55 spite these setbacks”-          could you discuss a bit more how the drugs that have passed pre clinical trials have overcome these obstacles, what are their bioavailability, toxicity or how were these deficiencies addressed?

We have added a few sentences and two additional references to explain why current TKI have better affinity/specificty, fewer off target effect and less severe and easily manageable side effects that we detailed.

First letter of  drug brand name should be capitalized, please go through the paper

Correction has been made

Line 94- it should be “of note”, instead of “of not”, you are missing the “e”

Correction has been made

Line 95- It should be “receive” NOT “received”. Sentence should be, “patients were not allowed to receive

Could you also define or describe what  standard will be considered a “fit” patient.

Correction has been made

In line 205 “Quizartinib” first letter was capitalized but it line 187 it was started with a small letter. So there needs to be consistency. In my opinion “brand name” of drugs should be written as a proper noun and started with a capital letter. So you need to go over the paper and do that for all the brand names of the drugs written.

Correction has been made

Line 274 it should be  “association with” NOT “association to”. In a in sentence you say that things are “associated with”, or has an “association with”

Correction has been made

Line 278, patient treated “with”, preferable use “with” here instead of treated “by”. Same applies to line 282- patients were treated “with” a combination …

Correction has been made

Line 308-309 is not very clear, probably need to elaborate on that sentence.

Correction has been made

Line333 what is EBMT, at this point the reader would have forgotten what that acronym stands for.

Correction has been made

 Line 390- better to use “unanswered question” than “unsolved questions”

Correction has been made

Line 431- it should be ineligible “for” intensive…, not “to”. So change the “to” with a “for”

Correction has been made

Line 432 -  association of FLT3 TKI “to” demethylating, please change that “to” and use “with”

You say something is associated with, not associated to.

Correction has been made

Line 435 this sentence is not complete. Maybe say-  “A major breakthrough about these drugs is that they are associated with minimal adverse events…..”

Correction has been made

Line 438 instead of unsolved question either say “unresolved questions” or “unanswered questions”

Correction has been made